# RNA-Seq Analysis of Trans-Differentiated ARPE-19 Cells Transduced by AAV9-AIPL1 Vectors

**DOI:** 10.3390/ijms25010197

**Published:** 2023-12-22

**Authors:** Alima Galieva, Alexander Egorov, Alexander Malogolovkin, Andrew Brovin, Alexander Karabelsky

**Affiliations:** 1Gene Therapy Department, Science Center for Translational Medicine, Sirius University of Science and Technology, 354340 Sirius, Russia; alima.galieva@gmail.com (A.G.); malogolovkin.as@talantiuspeh.ru (A.M.); brovin.an@talantiuspeh.ru (A.B.); 2Molecular Virology Laboratory, First Moscow State Medical University (Sechenov University), 119991 Moscow, Russia

**Keywords:** AAV, AAV9, AIPL1, RNA-seq, ARPE-19, Leber congenital amaurosis, LCA4, gene therapy

## Abstract

Inherited retinal disorders (IRD) have become a primary focus of gene therapy research since the success of adeno-associated virus-based therapeutics (voretigene neparvovec-rzyl) for Leber congenital amaurosis type 2 (LCA2). Dozens of monogenic IRDs could be potentially treated with a similar approach using an adeno-associated virus (AAV) to transfer a functional gene into the retina. Here, we present the results of the design, production, and in vitro testing of the AAV serotype 9 (AAV9) vector carrying the codon-optimized (co) copy of aryl hydrocarbon receptor-interacting protein like-1 (*AIPL1*) as a possible treatment for LCA4. The pAAV-AIPL1co was able to successfully transduce retinal pigment epithelium cells (ARPE-19) and initiate the expression of human *AIPL1*. Intriguingly, cells transduced with AAV9-AIPL1co showed much less antiviral response than AAV9-AIPL1wt (wild-type *AIPL1*) transduced. RNA-sequencing (RNA-seq) analysis of trans-differentiated ARPE-19 cells transduced with AAV9-AIPL1co demonstrated significant differences in the expression of genes involved in the innate immune response. In contrast, AAV9-AIPL1wt induced the prominent activation of multiple interferon-stimulated genes. The key part of the possible regulatory molecular mechanism is the activation of dsRNA-responsive antiviral oligoadenylate synthetases, and a significant increase in the level of histone coding genes’ transcripts overrepresented in RNA-seq data (i.e., H1, H2A, H2B, H3, and H4). The RNA-seq data suggests that AAV9-AIPL1co exhibiting less immunogenicity than AAV9-AIPL1wt can be used for potency testing, using relevant animal models to develop future therapeutics for LCA4.

## 1. Introduction

Monogenic diseases of the retina are a group of about 300 isolated or syndromic diseases, heterogeneous in clinical manifestations, which are united by similar development mechanisms—damage to one of the links of phototransduction or damage to the integrity of photoreceptor/auxiliary cells. The frequency of occurrence of monogenic retinal diseases ranges from 1:5000 (Stargardt disease) to 1:100,000 (Usher syndrome, achromatopsia). To date, there is no pathogenetically directed effective treatment for this group of diseases in wide clinical practice; however, promising methods of treatment are at different stages of clinical and preclinical studies [1].

Leber congenital amaurosis (LCA), the most rapid and severe form of hereditary retinal dystrophy, is usually inherited in an autosomal recessive manner and is characterized by early visual loss, nystagmus, and an absent or severely reduced electroretinogram (ERG) [2]. To date, mutations in 26 different genes encoding proteins important in retinal development and physiological function cause clinically distinct types of LCA [3].

Given the unique features of the eye (accessible location, small, isolated structure, immune privilege), hereditary retinal dystrophies are one of the most attractive targets for gene therapy. In recent years, the number of clinical trials of gene therapy for hereditary retinal dystrophies has increased, culminating in the registration of the first gene therapy for the treatment—voretigene neparvovec-rzyl (Luxturna^®^)—a recombinant adeno-associated virus (AAV) expressing the *RPE65* gene for the treatment of Leber’s amaurosis type II (LCA II) [4].

Among the mutations causing other types of LCA, about 5–10% are in the *AIPL1* gene encoding aryl hydrocarbon receptor-interacting protein like-1 (LCA4). Numerous allelic variants in the highly polymorphic *AIPL1* gene are associated with a wide spectrum of inherited retinal diseases including severe autosomal recessive Leber congenital amaurosis and retinitis pigmentosa. The AIPL1 protein functions as a photoreceptor-specific co-chaperone interacting with HSP90 to facilitate the stable assembly of retinal cGMP phosphodiesterase 6 (PDE6). AIPL1 was shown to interact with farnesylated proteins via its FKBP domain and to enhance the efficiency of farnesylation, help target farnesylated proteins to the endoplasmic reticulum for further processing, protect farnesylated proteins from proteolysis in the cytosol, and chaperone farnesylated proteins to their target membranes [5]. The clinical manifestations caused by AIPL1 deficiency are in the LCA spectrum with a severe course. These severe symptoms are caused by extensive and irreversible degeneration of rod and cone photoreceptors critical for visual phototransduction, in which AIPL1 plays an indirect but important role in maintaining functional integrity [5].

In the framework of this study, we assessed the differential gene expression of the arising retinal pigment epithelial cell line (ARPE-19) after transduction with AAV9-AIPL1 vectors. To assess the influence of the transduction effect itself, a comparison was made with the transcriptome of the cell line transduced with AAV9-GFP, and to assess the possible influence of the AIPL1 gene expression level, expression cassettes with the wild-type AIPL1 gene and a codon-optimized variant providing a higher level of transgene expression were used. Studying the effect of viral transduction on the gene expression profile may be useful, both in terms of predicting possible side effects from gene therapy, including immune response, and in terms of a more detailed study of the AIPL1 co-chaperone function. Thus, previously, the possibility was reported of restoring photoreceptor homeostasis in the case of PDE6-associated retinitis using an AAV-based gene addition technique [6]. Therefore, the study of possible pleiotropic effects from the overexpression of AIPL1, which is involved in the maturation of a whole range of proteins in photoreceptor cells, may also help open up new therapeutic niches for developing gene therapy products.

## 2. Results

### 2.1. ARPE-19 Cells Trans-Differentiation

Previous studies have already shown that ARPE-19 cells can trans-differentiate into neuron-like retinal cells under certain stimuli. For example, supplementation with the synthetic retinoid fenretinide [7,8,9] or bFGF [10,11] induced the acquisition of a neuron-like phenotype. Additionally, the trans-differentiation of retinal pigment epithelial cells into photoreceptors and retinal ganglion cells (RGCs) using viral transduction with neurogenin 2 has been described [12]. However, it should be noted that the complete differentiation of ARPE-19 cells into photoreceptors has not been demonstrated. In this study, we demonstrate that ARPE-19 cells, under treatment with fenretinide, preferentially express transcripts characteristic for cells of neuroretinal origin. Since our primary goal was to determine the most suitable method for the trans-differentiation of ARPE-19 cells, we started with the following experiment. To compare gene expression after different treatments, we cultured cells in the presence of 3 µM fenretinide and in the presence of basic fibroblast growth factor (bFGF) at a 20 ng/mL concentration under similar culture conditions. We expected to discover changes in expression underlying the effects of treatment.

We suggest that the low amplification signals of photoreceptor-specific genes *AIPL1*, *LCA5*, *ND4*, *NUB1*, *PDE6B*, and *RDH12* indicate that the cells did not acquire the basic properties of photoreceptors. However, the expression levels of the opsin 3 (*OPN3*) and *KLF4* genes could function as effective markers for determining the differentiation of retinal cells. *OPN3* is a non-visual opsin, called encephalospin/panaopsin for being expressed ubiquitously throughout the nerve system and epithelium [13]. The expression level of *OPN3* increases significantly in the inner retinal cells of developing retina at the different embryonic stages. *KLF4*, a transcription factor, has been implicated in neuronal development and differentiation. Its upregulation can promote neural differentiation and the acquisition of neuronal characteristics [14]. *TUBB3*, also known as beta-III tubulin, is considered to be a marker commonly associated with neuronal differentiation. Its increased expression indicates the maturation of neuronal cells. On the other hand, the decreased expression of *OPN3* may indicate a shift away from a neuronal fate. However, *OPN3* is typically expressed in retina, yet rod and cone photoreceptor cells have a lower expression than RGC [15]. Thus, the decrease we observed in *OPN3* expression level suggests differentiation of neuroretinal cells towards photoreceptors, while an induction of *KLF4* expression determines the cell types that give rise to late retinal progenitor cells, leading to an increase in RGC marker content. In summary, the increased expression of *TUBB3* and *KLF4*, along with the decreased expression of *OPN3*, may indicate a differentiation process, in the course of which ARPE-19 cells undergo differentiation towards a neurogenic phenotype distinct from retinal ganglion cells (Figure 1). It is important to consider these findings in the context of other markers and factors involved in the differentiation process to gain a comprehensive understanding of their specific cellular fate and differentiation pathway.

Thus, the literature data [9,11] was confirmed for markers *OPN3* and *KLF4* for directed differentiation, and the method for trans-differentiation with fenretinide was approved for our second experiment. On the second run, treated-with-fenretinide and non-treated cells were tested for the expression of all genes listed in Appendix A.

Most of the data on differentiation markers has not been updated since 2017. Therefore, the use of modern transcriptomic analysis methods will allow us to describe the trans-differentiation process in more detail, as has been done in article [16]. The following genetic markers were previously considered characteristic of specific cell types: pigment epithelium, *RPE65*, *RRH*, and *RDH11* [8]; photoreceptor markers, *CRX*, *IRBP*, *RRG*, and *SAG* [9]; of neuronal and glial cells of the retina, *NF-M*, *NF-H*, *NSE*, *NRL*, *OPN3*, *OPN4*, *SYP*, and *TUBB3*, [9,11]; and pluripotency markers and transcription factors, *OCT4*, *SOX4*, *NANOG*, *KLF4*, *PAX6*, and *MITF* [11]. In our work, in addition to already studied differentiation markers, we also examined the expression of genes that potentially could be co-expressed with *AIPL1*, since it is our gene of interest. Drawing on data from the STRING database (Figure 2), the genes *LCA5*, *ND4*, *AHR*, *NUB1*, *PDE6B*, *RDH12*, *GUCY2D*, *TULP1*, and *CRB1*, which are also involved in the cascade of the phototransduction cycle, were also considered in the study.

### 2.2. AIPL1co and AIPLwt Expressed in HEK-293 Cells

In order to confirm the functionality of the generated viral vectors, HEK-293T were transduced by AAV9-AIPL1co and AAV-AIPL1wt with various MOI (Appendix A). AIPL1 mRNA transcripts were detected at 48 h post transduction and analyzed with a comparative RT-qPCR. AAV9-AIPL1co demonstrated a higher transduction and expression efficiency than AAV9-AIPLwt based on the *AIPL1* transcript copies detected at a different MOI range. *AIPL1* gene product was detected with Western blot using HEK-293T cells transfected with pAAV-AIPL1co-His and pAAV-AIPL1wt-His plasmids. An AIPL1 protein-specific band at 36 kDa was detected in cell lysates with anti-His antibodies. Relative optical density units (RDU) of the protein-specific bands were calculated and plotted in a graph to visualize the difference between pAAV-AIPL1co-His and pAAV-AIPL1wt-His transfected cells. AAV-AIPL1co-His showed a 1.39 times increase in overall protein product compared to pAAV-AIPL1wt-His transfected HEK-293T cells.

### 2.3. qPCR Screening

The STRING database [17] search performed to determine genes involved in AIPL1-associated biomolecular processes allowed us to generate a gene list. The list included genes associated with different forms of Leber congenital amaurosis, involved in disruptions in phototransduction (*AIPL1*), retinoid cycle (*RDH12, RPE65*), and transport across the photoreceptor connecting cilium (LCA5), as well as genes evidently interacting with AIPL1 (*NUB1, AHR*).

Secondly, qPCR screening was used for determining quantitative and qualitative information about these genes. As a result of the screening, we obtained data on changes in the expression level of genes of interest *KLF4*, *TUBB3*, *RPE65*, *OPN3*, *AIPL1*, *LCA5*, *ND4*, *AHR*, *NUB1*, *PDE6B*, *RDH12*, and *ARR3*, and housekeeping gene *PPIA*. In the results of the primary analysis, test systems for *AIPL1*, *LCA5*, *ND4*, *NUB1*, *PDE6B*, *RDH12*, *KLF4*, *TUBB3*, and *OPN3* were selected, the relative expression in trans-differentiated ARPE-19 cells of which is presented in Figure 3. We observed a significant increase in the expression of the *TUBB3*, *RPE65*, *ND4*, *KLF4* and *AIPL1* genes and a decrease in the expression of *RDH12*, *PDE6B*, *LCA5*, *AHR*, *NUB1*, and *ARR3*. This data highlights the enhancement of neuronal features and photoreceptor functions in fenretinide-treated cells.

Based on the results of screening the level of expression of various genes, it can be concluded that exposure to fenretinide directs cell differentiation along the neuronal pathway, but their properties are not sufficient to express genes involved in the cycle of visual phototransduction. To more effectively target photoreceptor genes, it is necessary to use a more specific cell line as an optimal model for in vitro tests. The most representative systems are for *AIPL1*, *ND4*, *PDE6B*, and *RDH12*, since the endogenous level of these genes is sufficient to study the level of relative changes in expression levels. For other targets, it is recommended to use a more specific cell line close to the photoreceptors, or to use other, more specific markers for the ARPE-19 cells used.

### 2.4. RNA-Seq Data Analysis

Genetic profiling using RT-qPCR for the abovementioned genes confirmed the transitioned phenotype. Then, differentiated ARPE-19 were transduced with 100,000 MOI AAV vectors carrying transgenes (Figure 4). Non-transduced cells were used as a control. After 24 h, cells were harvested, and total RNA was extracted to perform RT-qPCR and RNA-seq analysis. After the isolation of total RNA, the RNA integrity number was 9.5–10, which is a high-quality value.

A summarized reads quality control report is presented in the Appendix A. As a result of reads quality control with the FastQC tool and the summarizing of reports with MultiQC, the raw reads of all the samples were good quality with a per base Phred score of 35.73 and higher (which corresponds to a base call accuracy higher than 99.9%) [18]. Per base sequence content was slightly biased for the first 12 nucleotides over all samples, which is common for cDNA libraries prepared with a random primers mix for reverse transcription due to the not-absolute randomness of the primers and bias in their hybridization during amplification steps. Per base N content was 0.05% for first base, 0.01% for second base, and 0% for the rest bases, which is a sign of high quality of base calling. Sequence length over all reads, over all samples, was identical and equaled 60 base pairs, which correlated with what was expected. The sequence duplication level was between 100 and 500 for 11.2% of reads of only one sample (AAV9-AIPL1wt) (which was signed by FastQC algorithm as “failed”); for the rest of the samples, this value was lower and part of unique reads was 40% and more; the peak of duplicated sequences might be a signal of enrichment bias of residual rRNA, so a slight rRNA sequences presence in libraries was acceptable for subsequent analysis. The presence of overrepresented sequences in reads over all samples was 0.24% or less, and all of them were recognized as Illumina adapters; a small amount of adapters’ sequences was acceptable for subsequent mapping by STAR due to the peculiarity of the STAR algorithm, so the trimming of residual technical sequences was not essential.

GC content over all reads of all the samples was unimodal, with a mode of about 51–54%, which does not correlate with the literature data [19]. As previously reported, GC content might vary between replicate lanes as a result of differences in sequencing depth, i.e., the total number of reads produced in a given lane [20], and also due to PCR bias [21], which might explain this inconsistency. It is worth noting that only three non-transduced samples had the nearest to previously detailed data GC content (51%) and all transduced samples had higher GC content (52–54%). The observed phenomena might be explained by the activation of specific genome regions called ridges [22], but it requires further investigation.

As a result of summarizing alignment statistics (Appendix A), at least 74.8% of reads mapped uniquely; from 13.2% to 24.3% of reads mapped to multiple loci; 0.2% of reads over all samples mapped to too many loci; and less than 0.7% were unmapped.

As a result of alignment quality control (Appendix A), from 61.9% to 66.1% mapped to CDS exons regions, from 7.5% to 10%—to introns, from 4.4% to 5.7%—to 5′-UTRs, and from 17.4% to 19.8% to 3′-UTRs. Gene body coverage was almost uniform, with slight skewness from −4.9 to −4.3, and all samples showed little or no bias. Read duplication level over all aligned reads was acceptable (relatively small areas under the curves), i.e., most reads of all the samples had a low number of exact duplicates. Splice junction curves for all the samples were not saturated, which is sign of not enough sequencing depth for alternative splicing analysis, as using an unsaturated sequencing depth would miss many rare splice junctions [23], which still might have been suitable for our research, as we were performing differential gene expression analysis, not splice isoforms. However, it has been reported that adding more sequencing depth after 10 million reads gives diminishing returns on the power to detect differentially expressed genes, whereas adding biological replicates improves power significantly, regardless of sequencing depth [24]. This fact should be considered when designing RNA-sequencing experiments. The resulting output table from HTSeq-count, with summarized reads per gene, is presented in the Appendix A.

Sample-level quality control was performed with principal component analysis and the hierarchical clustering method. Both the PCA plot and the clustering heatmap were generated using functions of DEseq2 [25] and pheatmap [26] R libraries. With the quality control methods, it was estimated that one non-transduced control (CC1) sample lay out of the cluster of control replicates, while the rest of the samples demonstrated the expected grouping among replicates, within sample types and sample groups spread across the two PCs. PC1 accounts for 59% of the variance, and PC2 accounts for an additional 19% (Figure 5). The heatmap confirmed that the majority of the variance within the dataset was described by the first two PCs, as replicates clustered together as a block for each sample group, except the CC1 sample. Intergroup variability describes technical or biological variability, which characterizes whether experimental conditions represent the major source of variation in the data [27]. Outliers would indicate an overall effect of experimental covariates and batch effects, and due to the high dissimilarity of the CC1 sample, it has been decided to remove it from the dataset.

In total, 62,757 genes were annotated in the human reference genome. Mapping and reads per gene summarizing resulted in 30,339 genes with zero counts for all sample replicates. We found that the number of the differentially expressed genes with a 3-fold change (Log2Fold change = 1.58) did not significantly vary with a decrease in the *p*-value adjusted threshold (Appendix A). This means that the use of more stringent parameters will not lead to the elimination of less significant differentially expressed genes (DEGs), but to a decrease in the total number of detected DEGs. Setting these thresholds, we obtained sets of DEGs for different pairs of comparison, the number of which is presented in Table 1 and visualized in volcano plots (Figure 6 and Figure 7). According to GeneOntology, each set of DEGs, separated to up- and downregulated subsets, are distributed by biological processes described in Figure 8.

### 2.5. Genes Relevant for Both AAV9-AIPL1wt and AAV9-AIPL1co Transduced Cells

Data obtained after transcriptome sequencing showed that the neuroretinal cells that we worked with obtained new features as a result of induced trans-differentiation. Therefore, we checked the differential expression of genes relevant for sensory disorders after analyzing the Venn diagrams in Figure 9.

The eight most relevant genes were identified: *MAF*, *ABCB5*, *GAS1*, *WHRL*, *UAP1L*, *ANGPTL2*, *UCN* upregulated, and the *RRM2* gene downregulated (Figure 10).

### 2.6. Genes Overexpressed in ARPE-19 Cells Transduced with AAV9-AIPL1co

The volcano plot in Figure 6 represents differentially expressed genes in cells transduced by AAV9-AIPL1co (codon-optimized) versus AAV9-AIPL1wt (native sequence). According to the data presented in the volcano plots, the most differentially expressed genes in cells transduced with AAV9-AIPL1co, apart from *ABCB5*, were *IL1RN* and *PPARGC1A*.

The *PPARGC1A* gene encodes PGC1-α, the main PPARɣ (peroxisome-proliferator-activated receptor-γ) transcription factor coactivator. It provides a connection between external stimuli and the processes of mitochondrial biogenesis. PGC1-α is known to be a key regulator of cell metabolism, in particular by controlling the metabolic status of adipose tissue and the type of muscle tissue fibers.

It has been previously reported that an overexpression of PGC-1α and induction of PGC-1α through fasting, physical exercise, glucagon, or an elevated level of cAMP is associated with increased *IL1RN* mRNA [28]. We also confirm this observation with RNA-sequencing data, in which we found a significant increase (Figure 11) in IL1RN transcripts numbers in the results of both AAV9-AIPL1wt transduction (3-fold) and AAV9-AIPL1co transduction (16-fold) in comparison with non-treated control.

Moreover, we observe the remarkable change in expression of other related-to-IL-1 genes as well (Figure 12). Most of the IL-1 family genes are located at a single locus on chromosome 2. Therefore, the corresponding alteration of the collocated genes is possible due to the close proximity of regulatory regions within the gene cluster [29].

In a nutshell, the overexpression of two genes (*IL1RN* and *PPARGC1A*) may significantly influence both cellular immune response and transcription regulation. We explain in detail the character of the changes observed in the Section 3.

### 2.7. Genes Overexpressed in ARPE-19 Cells Transduced with AAV9-AIPL1wt

The main transcription factor controlling gene induction in response to the pro-inflammatory signaling of IL-1β is NF-κB. The RNA-Seq data revealed the general upregulation of NF-κB-associated genes (Figure 13) in cells transduced by AAV9-AIPL1wt, but not the cells transduced with AAV9-AIPL1co. Instead, they were found to have comparably low levels of the same set of NF-κB-associated genes.

In contrast to the AAV9-AIPL1co data, among the most differentially expressed genes in cells transduced with AAV9-AIPL1wt were *IFIT1*, *ISG15*, and *IFITM10*, the expression of which increased remarkably (Figure 6). These are the members of interferon-stimulated pathways, including some transcription factors responsible for the implementation of interferon response.

We found that AIPL1 overexpression in both AAV9-AIPL1wt and AAV9-AIPL1co transduced cells had comparable effects on the expression of its interactors (Figure 14). A number of AIPL1 interactors—POT1, TINF2, and ACD—form an integral part of a shelterin complex. In generated heatmaps (Figure 15) we discovered changes suggesting possible involvement of AIPL1 in shelterin complex regulation. Section 3 discloses our view on the relevance of these data for the global control of histone gene expression (Figure 16).

The unusual response of histone gene expression to transduction with adeno-associated viral vectors may contribute to the difference in intracellular innate immune response. The mRNAs attributed to a few interferon-stimulated genes (ISGs) are present among differentially expressed genes, according to our volcano plots (Figure 6). We checked for changes in a comprehensive range of interferon-related genes, and found them substantially increased after transduction with AAV9-AIPL1wt (Figure 17). On the contrary, AAV9-AIPL1co transduced cells demonstrated relatively low ISG induction.

2′-5′ oligoadenylate synthetases (OAS) are interferon-induced antiviral enzymes serving as the major components of the intracellular antiviral system. We found *OAS1* to be one of the most differentially expressed in AAV9-AIPL1wt versus both non-transduced cells and AAV9-GFP control (Figure 6B,D). The expression of the full set (*OAS1*, *OAS2*, *OAS3,* and *OAS*), as well as the gene of the mitochondrial antiviral protein *MAVS*, was significantly altered after transduction with AAV9-AIPL1wt (Figure 18).

We also observe unique expression patterns of *H2A.Z* genes (Figure 19), which are variants of histone 2A shown to play a meaningful role in the control of ISGs transcription during viral invasion [30].

A general reflection drawn from these findings in our RNA-Seq was the ability of AAVs coding for codon-optimized *AIPL1*, despite many similarities, to stimulate specific expression, along with decreased interferon-stimulated genes expression. In cells transduced with AAV9-AIPL1wt, the interferon-stimulated genes are overexpressed. Several mechanisms are hypothesized as a major explanation for the observed phenomena in the Discussion.

## 3. Discussion

The retina is a complex structure comprising multiple layers of distinct neuronal cells that are difficult to access in vivo due to its location in the inner part of the eye. The human retinal pigment epithelium (RPE) is a monolayer of cells that supports the normal functioning of photoreceptor cells in the retina. In the pathogenesis of inherited retinal disorders, RPE cells often play a critical role. All experiments were performed on ARPE-19 cells. ARPE-19 is a cell line of a spontaneously arising human RPE with a normal karyotype, obtained after selective trypsinization of a primary RPE derived from the normal eyes of a 19-year-old male individual [31]. It has been shown that ARPE-19 is a population of uniform epithelial cells with strong growth potential, still exhibiting a tendency to senesce in culture, which demonstrates its non-transformed condition.

The characteristic phenotype of ARPE-19 cells is not identical to photoreceptors, which in their own nature exhibit neuroretinal features. Neuroretinal cells, in contrast to RPE cells, form the inner photo-sensitive layer of the eye. The neuroretina consists of six types of cells of neuronal origin (two types of photoreceptors—cones and rods, horizontal, bipolar, amacrine, and ganglion cells) and three types of glial cells (Müller glial cells, astrocytes, and microglial cells) [32]. Thus, with trans-differentiation, we made an attempt to create a cell model with the desired phenotype.

The set of genes we found to be significantly altered after transduction with both wild-type and codon-optimized AAV9-AIPL1, according to RNA-Seq, includes the *MAF*, *ABCB5*, *GAS1*, *WHRL*, *UAP1L*, *ANGPTL2*, *UCN*, and *RRM2* gene. We observe that the alteration in expression of these genes is associated with AIPL1 and propose their function on the basis of previous data.

MAF is a transcription factor which plays a significant role during eye development, as was proved by numerous ocular developmental abnormalities associated with *MAF* mutation in the human genome [33,34]. However, the exact role of MAF in the physiological processes in the retina remains unclear; it may be involved in the maintenance of visual function by protection from oxidative stress [35].

*ABCB5* is a marker for adult limbal stem cells of the cornea [36]. ABCB5+ cells can regenerate a cornea in a mouse with limbal stem cell deficiency (disease of the corneal limbus causing blindness), suggesting a therapeutic potential of this cell population for treating some types of blindness [37]. Growth arrest specific gene 1 (*GAS1*) has been reported to inhibit cell cycle progression in vitro [38].

It has also been demonstrated that the expression pattern of *GAS1* in the eye inhibits RPE proliferation. In transgenic mice carrying a targeted mutation in the *GAS1* locus, microphthalmia developed: it was histologically revealed that the remnant mutant eyes were ingressed from the surface with minimal RPE and lens, and disorganized eyelid, cornea, and neural retina [39]. An alteration in *GAS1* has been revealed for patients with keratoconus (a condition where the cornea becomes progressively thin and protrudes conically) [40]. Keratoconus leads to astigmatism, myopia, and corneal scarring, with eventual loss of vision. The major form of keratoconus is asyndromic, in which the cornea alone is affected. But syndromic types exist as well, and are associated with Down, Leber congenital amaurosis, Turner, Marfan, and Ehlers–Danlos syndrome [41].

Mutations in the *WHRN* gene encoding whirlin, a PDZ domain molecule involved in stereocilia elongation, cause deafness, in particular in Usher syndrome type 2 [42]. Meanwhile, urocortin (*UCN*), which encodes a peptide from the corticotropin-releasing factor family, protects against retinal degeneration, and it has been shown to be expressed in post-ONC in the two sustained αRGC types, especially in response to AAV, according to single-cell RNA-sequencing [43]. Angptl2 has a role in acute inflammation in the eye [44].

To sum up, we observe upregulation of the set of genes associated with inherited diseases of eye development after AAV-AIPL1 transduction. Specifically, the expression of these genes markedly increased after transduction with the adeno-associated viral vector encoding for the codon-optimized *AIPL1* sequence. This confirms the higher specificity of action for AAV9-AIPLco (codon-optimized).

Regarding the exceptionally high levels of transcripts of *PPARGC1A* and *IL1RN* detected, especially for AAV9-AIPL1co samples, we anticipate the need for an in-depth study. We suppose that corresponding upregulation of both *PPARGC1A* and *IL1RN* occurs due to AIPL1-specific activity and not adeno-associated viral vectors transduction, as in cells transduced with AAV9-GFP there is no such alteration detected.

The *PPARGC1A* gene encodes for PGC1-ɑ, the main PPARɣ (peroxisome-proliferator-activated receptor-γ) transcription factor coactivator. It provides a connection between external stimuli and the processes of mitochondrial biogenesis. PGC1-ɑ is known to be a key regulator of cell metabolism, in particular by controlling the metabolic status of adipose tissue and the type of muscle tissue fibers. As PPARGC1A regulates the activity of a set of nuclear receptors and controls mitochondrial function, the elevated level of its expression may explain the eerie correlation of a wide array of genes detected. Specifically, it was previously reported that overexpression of PGC-1α and induction of PGC-1α by fasting, physical exercise, glucagon, or an elevated level of cAMP was associated with increased *IL1RN* mRNA [28]. We confirm this observation using the RNA-seq data.

The elevated level of *IL1RN* gene expression that was detected in our data might be able to flip the script. The anti-inflammatory antagonist Il1Ra, that the *IL1RN* gene encodes for, belongs to the interleukin-1 family and is able to modulate immune response by blocking interleukin-1 receptors. According to RNA-seq counts, a significant change (Figure 11) in the *IL1RN* transcripts number occurs in the results of AAV9-AIPL1wt transduction (3-fold in comparison with non-treated control), as well as AAV9-AIPL1co transduction (16-fold).

The overexpression of the *IL1RN* gene in cells transduced with AAV9-AIPL1co potentially has implications for local immunity, including the response to pro-inflammatory cytokines, interferon response, and Toll-like receptor (TLR) activation. IL-1Ra is widely known for its ability to suppress the activity of interleukin-1, a pro-inflammatory cytokine involved in the immune response, by selective binding of its receptor [45].

Obviously, our immune system is a complex and interconnected network, and the effects of *IL1RN* overexpression on cellular immune responses would likely involve interactions with multiple signaling pathways, cytokines, and various cellular processes.

Firstly, along with the genes of the IL-1 cluster, we checked the gene expression of the IL-1 receptor accessory protein (*IL1RAPL1*), an essential component of IL-1 signaling, and its level was altered coordinately with the IL-1 receptor gene. In its entirety, the data we show testifies that the IL-1 signaling pathway is inhibited in cells transduced with AAV9-AIPL1co. As the stimulation of the IL-1 receptor induces IL-6 gene expression [44], we checked for gene expression and found that the IL-6 level was reduced in AAV9-AIPL1co samples compared with AAV9-AIPL1wt, nearly the same as in the non-transduced control.

However, the relationship between IL-1Ra and other components of the immune system is complex, and it is important to consider various factors when evaluating the potential impact of *IL1RN* overexpression on immune responses. For instance, Toll-like receptors (TLR), a crucial component of the innate immune system, recognize various pathogen-associated molecular patterns and initiate immune responses. It was demonstrated earlier that IL-1Ra inhibits TLR signaling pathways and the subsequent immune responses elicited by *TLR9* activation [46]. Both interleukin-1 and Toll-like receptor pathways result in NF-κB transcriptional activation. We thoroughly investigated the data, and demonstrated the general upregulation of NF-κB-associated genes (Figure 13) in the mRNA of cells transduced with AAV9-AIPL1wt. Meanwhile, the cells transduced with AAV9-AIPL1co, compared to the wild type, were found to have significantly lower levels among the same set of NF-κB-associated genes. Further research is needed to fully understand the potential effects of *IL1RN* overexpression and its impact on the overall immune response, including responses to interferons and TLR activation, and the role of AIPL1 in regulation of IL1RN activity.

The interplay between interferons and IL-1Ra, including the activation of the least in response to IFNs [47,48], could potentially lead to alterations in immune signaling.

The induction of *IL1RN* gene expression, besides the possible participation of PGC1-ɑ, has extra arguments, including AIPL1 protein interactions. We searched for AIPL1 interactors at the BioGRID database that collects genetic and protein interactions, and generated a heatmap based on our RNA-seq data for the list of interactors. In the results, we found that AIPL1 overexpression has ambivalent effects on the expression of its interactors (Figure 14).

There is a set of genes present in the STRING network as a separate cluster (Figure 2), found in a lower part of the heatmap from which we highlighted AIPL1 interactors POT1, TINF2, and ACD. These genes code for members of the shelterin complex, and their interactions with AIPL1 were confirmed by mass-spectrometry data [49]. For these genes, there was a regularity in the heatmaps pattern revealed (Figure 15), which is possible evidence of AIPL1 involvement in shelterin complex regulation. The role of the shelterin complex, specifically its member RAP1, in global control of histone gene expression was confirmed in several studies [50,51].

We also found an unusual response of histone gene expression to transduction with adeno-associated viral vectors: the expression of histone genes was markedly increased in transduced samples (Figure 16). This fact was also previously noted in transcriptomic studies following adenovirus infection [52], but the exact molecular mechanism of such upregulation remained unclear. During the late phase of adenovirus infection (30 h past), almost half of the upregulated genes were involved in nucleic acid metabolism (specifically, *HIST1H2BC* and *HIST1H2BF*). The possible mechanism underlying the effect of the upregulation of histones in response to interferon regulatory factors activation was investigated earlier [53]. Interestingly, not only could interferon response stimulate the expression of histone genes, but the depletion of histones was able to activate interferon regulatory factors [54].

Another reason for the increase in histone-attributed counts is their possible overrepresentation due to a change in structure. We propose that, although most histone transcripts are known to be not polyadenylated, as we used oligodT for library preparation, the histone transcripts were polyadenylated. The notoriety of this matter of fact was possible due to the study of polyadenylation of the human histone *H2B* gene transcripts [55]. In this study, it was reported that, under certain conditions, in particular, cellular differentiation or distress, histone transcripts could be polyadenylated. We, instead, observe the presence of such polyadenylated histone mRNA transcripts enriched after viral transduction, as we are able to distinguish and attribute these counts in RNA-seq data.

It was speculated earlier that the interferon response could be possibly triggered by the expression of noncoding RNA generated from heterochromatic repeats or endogenous retroviruses [52]. However, several studies have shown that the activation of histone genes is associated with the production of interferons. Interferon-stimulated genes (ISGs) can directly regulate the expression of histone genes [53,56]. We found in our RNA-seq that, after AAVs transduction with AAV9-AIPL1co, the ISG expression decreased. In contrast, in cells transduced with AAV9-AIPL1wt, there is noticeable increase in the expression of interferon-stimulated genes (ISGs), such as *OAS1*, *IFIT1*, *ISG15*, and *IFITM10* (volcano plots in Figure 6). In order to test the hypothesis that the ISG levels altered after transduction, we checked for the change in expression of a redundant set of interferon-stimulated genes (Figure 17), and found a general increase in transcription of the ISGs after transduction with AAV9-AIPL1wt and a comparative decrease after transduction with AAV9-AIPL1co.

There were notably decreased genes responsible for innate immune response to the virus (*OAS1*, *IFIT1*, *ISG15*, *IFI6*) in AAV9-AIPL1co in comparison to the virus encoding the wild-type AIPL1 sequence. This in turn, theoretically, is reasonable due to the abovementioned connection of the histone-related response to interferon.

One of the major components of the intracellular antiviral system, *OAS1* is not ‘yet another interferon-responsive gene’. It belongs to the family of genes coding for enzymes 2′-5′ Oligoadenylate synthetases (OAS): *OAS1*, *OAS2*, *OAS3,* and *OASL*. All four genes are clustered at human chromosome 12q24.1 and *OASL* at 12q24.2, and all are IFN-inducible, but OASL is induced directly by viral infection. These enzymes require dsRNA for their catalytic activity. Thus, usually the OAS1 function is associated with restriction of RNA virus infection and progression [57]. Some of these viruses have developed the system of 2′,5′-phosphodiesterases that antagonizes OAS proteins [58]. It has been reported that AAV transduction may induce dsRNA-mediated innate immune response activation in a cell type-specific and transgene-dependent manner [59]. In our experiment, we see coordinated overexpression of *OAS1* (Figure 18A) and other dsRNA- responsive genes’ (Figure 18B) ISGs in AAV9-AIPL1wt samples, which characterizes the elevated responsiveness of these cells. It is quite evident that structural features of AAV single-stranded or self-complementary DNA could have a determining influence on short RNAs generation.

One of the possible cellular mechanisms to explain the rise of ISGs in cells transduced with AAV9-AIPL1wt was described earlier [30]. *H2A.Z* is a variant of histone 2A involved in several processes such as transcriptional control, especially for interferon-stimulated genes. When *H2A* is present within the bodies of ISGs, it is absent from the IFN-stimulated response elements’ proximal promoters and replaced with histone variant *H2A.Z*. Preliminary interferon stimulation leads to the loss of *H2A.Z* from interferon-responsive sequences, establishing transcriptional memory and creating a more potent antiviral state leading to robust inhibition of virus replication [30]. *H2A.Z*-containing nucleosomes seem to play a significant role in the regulation of ISGs transcription. It was demonstrated that shRNA-mediated knockdown of *H2A.Z* resulted in increased mRNA production of ISGs, and created a more potent antiviral state, leading to more robust inhibition of virus replication [30].

This suggests that histones can dramatically influence the interferon response. Furthermore, histones can act as DAMPs (damage-associated molecular patterns) and trigger the activation of immune cells, including the production of interferons [60,61]. The process of virus-induced inflammation, as we see, can stimulate the expression of histone genes, which by itself contributes to the regulation of immune responses and inflammation. Moreover, the connection between histone gene activation and the interferon response has a reciprocal relationship: as the response to interferon can influence the expression of histone genes, and histones, in turn, can modulate the interferon response.

Finally, the release of histones can have toxic effects on cells and stimulate apoptosis. Histones are highly basic proteins; this ensures high affinity to DNA. In the result of certain processes, such as inflammation and cell death, histones can be released from cells and enter the extracellular space. When histones are released in excessive amounts, they can trigger a series of detrimental effects on cells. Histones can disrupt cell membranes, cause oxidative stress, and activate intracellular signaling pathways that promote cell death. Excessive histone release has been shown to induce the production of interleukin-8 in ARPE-19 cells, and at concentrations more than 20 μg/mL is toxic to cells [62].

Interestingly, in each sample sequenced, we observed a unique pattern of histone transcripts expression. Some of the genes that were upregulated belong to the same histone gene cluster (Figure 16), i.e., collocated at the same chromosome, and possibly co-regulated. The expression of *H2A.Z* in our samples correlates to the effects observed. The higher level of *H2A.Z* transcripts was detected in cells transduced with AAV9-AIPL1co (Figure 19).

There is evidence to suggest that chaperones and co-chaperones can modulate the innate immune response and influence the cellular response to interferons. Few studies have indicated that chaperones and co-chaperones can modulate the innate immune response by regulating the activity of key signaling molecules involved in the response to interferons and other immune stimuli [62].

We observe the interferon response both to the AAV9-AIPL1wt and AAV9-AIPL1co, but in the case of codon-optimized samples, it could be compensated for with the specific activity of our gene of interest. Obviously, if there exists a feedback loop regulating the wild type, it does not pertain well to codon-optimized *AIPL1*. It is possible that the difference in mRNA structure does not allow effective control of codon-optimized *AIPL1* transcripts.

Data that were generated as a result of this study show innate immunity activation, specifically, interferon-stimulated genes induction after AAV transduction. There was no observational study of such depth performed before for innate immune response to AAV, which occurs in limited cell lines. Comprehensive analysis provides valuable insights into the innate immune response triggered by AAV gene therapy. We pay specific attention to intracellular immune response, including activation of histone genes transcription and dsRNA innate immune response activation, because these are the main pathways mediating immune reactivity in retina.

Previously, it was reported that the effects of gene therapy progressively improved, reached a peak, and then declined during clinical trials in patients with Leber’s congenital amaurosis after subretinal administration of AAV vectors encoding therapeutic transgenes [63]. According to preclinical experiments, the use of gene therapy can substantially improve the visual function but does not slow the process of retinal degeneration, due to the finding of progressive retinal thinning after AAV transduction [64]. Until now, it remains unknown if the activation of the innate immune response in the eye after AAV gene therapy is able to influence the expression of transgenes in patients with congenital retinal diseases. Our findings confirm and extend previous studies, highlighting the significance of the innate immune response, specifically dsRNA-mediated, and the importance of preclinical AAV evaluation in different cell lines and organoids. The most encouraging results from this study demonstrate that codon optimization and rational design is able to transcend limitations and reduce the innate immune response to an acceptable level. For some reason, AAV containing our patented codon-optimized sequence of *AIPL1* does not stimulate the interferon response and exhibits the immunosuppressive properties which makes this vector attractive from a perspective of development and gene therapy applications. However, the exact mechanism of this effect should be further studied in depth.

## 4. Materials and Methods

### 4.1. Genes and Plasmids

Protein coding sequence of human AIPL1 (AF148864.2) was retrieved from GenBank. Codon optimization was performed using EMBOSS backtranseq and verified with the Graphical Codon Usage Analyser [65]. AIPL1 codon-optimized (Appendix A) and wild-type sequences were synthesized by Topgenetech (Montréal, QC, Canada) and then recloned into pAAV-MCS (Cat. number VPK-400, Cell Biolabs, San Diego, CA, USA) to produce pAAV-AIPL1co and pAAV-AIPL1wt. In addition, AIPL1 sequences were fused with His-tag and also cloned in pAAV-MCS to generate pAAV-AIPL1co-His and pAAV-AIPL1wt-His. The pAAV-GFP (Cat. number VP-401, Cell Biolabs, USA) was used as a transfection control. The pAAV-RC2/9 (AddGene, 112865, Watertown, MA, USA) and pHelper (Part No. 340202, Cell Biolabs Inc., San Diego, CA, USA) with pAAV-AIPL1co and pAAVAIPL1wt, were used to produce AAV vectors.

### 4.2. AAV Production and Quality Control

Briefly, suspension-cultured human embryonic kidney cells (HEK-293) were transfected with a composition of pAAV-RC2/9, pHelper, and gene transfer (either pAAV-AIPL1co, pAAV-AIPL1wt or pAAV-GFP) vector with molar ratio 5:2:1, respectively, using transfection reagent PEI MAX (Linear polymer, MW 40,000, Polysciences Inc., Warrington, PA, USA) mixed with plasmid DNA in proportion 5:1. Recombinant AAVs were produced by cultivation in 250 mL Erlenmeyer flasks (Corning, Corning, NY, USA). After 96 h, the cells were harvested and lysed with 0.05% Tween-20 for 1 h, and additionally treated with benzonase (30U) for 2 h. Then, lysates were clarified using diatomaceous earth (HyFloSuperCel) by filtering the sample through a filter with a 0.22 μm pore size. Tangential filtration of the clarified lysates of AAV9-AIPL1co, AAV9-AIPL1wt, and AAV9-GFP was performed on a Labscale tangential flow filtration system using a 100 kDa cassette (Millipore Pellicon^®^ XL100, Merck, Rahway, NJ, USA). Chromatographic purification was carried out using the CaptureSelect POROS AAVX affinity resin (ThermoFischer Scientific, Waltham, MA, USA) with Bio-Rad Quest 10 Plus system. Finally, recombinant AAV vectors were concentrated to a volume of 1 mL through dialysis in a 1x phosphate-buffered saline (PBS) (0.37 M NaCl) using VivaFlow200 tangential filtration system (Sartorius, Bohemia, NY, USA), and 0.001% Pluronic F-68 was added (Sigma-Aldrich, Gillingham, UK). Purified viral preparation was subjected to analytical testing using quantitative polymerase chain reaction (qPCR) method (genome copies determination), low-angle dynamic light scattering (aggregates, viral particles concentration), size exclusion chromatography (aggregates, low molecular weight fractions), transmission electron microscopy (identity, purity, empty-to-full ratio), denaturing polyacrylamide gel electrophoresis (identity, purity).

### 4.3. RT-qPCR and Gene Expression Analysis

Quantitative reverse transcription polymerase chain reaction (RT-qPCR) was used to confirm the AIPL1co and AIPL1wt expression. Human AIPL1 gene specific primer pair universal for wild-type and optimized AIPL1 gene (Forward: 5′-GTGGCTGAAGCTGGAGAAG-3′, Reverse: 5′-CTCCAGCTCCAGCACTTTC-3′), with fluorescent probes specific for each gene (WT: 5′-FAM-AGTGCCTGCTGAAGAAGGAG-BHQ1-3′, OPT: 5′-FAM-CAACACCCTGATCCTGAAC-BHQ1-3′), was designed. The *PPIA* gene was used as a reference for gene expression normalization [66]. RT-qPCR was developed to quantify the level of AIPL1co and AIPL1wt mRNA in transfected or transduced cells. Statistical analysis of gene expression was performed with ddCT method [16]. Gene expression of photoreceptor-associated markers was also measured in ARPE-19 cells. Briefly, the ARPE-19 cells were incubated for 5 days at 37 °C with 5% CO_2_ without changing the medium. Lysis was carried out using the Lyra+ kit (Biolabmix, Novosibirsk Oblast, Russia) according to the manufacturer’s protocol for RNA isolation. cDNA synthesis was performed using the M-MuLV-RH RT kit (Biolabmix, Russia) and 2 μg of total RNA with polyA primers. Amplification of target genes (*PPIA*, *NUB1*, *AHR*, *LCA5*, *PDE6B*, *RDH12*, *ND4*, *KLF4*, *AIPL1*, *TUBB3*) was carried out with 100 ng of cDNA per reaction using the HS-qPCR Hi-ROX kit (Biolabmix, Russia) and specific primers (Appendix A). Additionally, expression of (*PPIA*, *OPN3*, *KLF4*, *TUBB3*) genes was measured in ARPE-19 cells treated with basic fibroblast growth factor (bFGF) at a concentration of 20 ng/mL for 5 days.

### 4.4. Western Blot

AIPL1co and AIPLwt protein expression was confirmed with Western blot. Briefly, HEK-293T cells were transfected with pAAV-AIPL1co-His and pAAV-AIPL1wt-His plasmids and Lipofectamine 3000 (ThermoFisher Scientific, Waltham, MA, USA). After 24 h post transfection, cells were harvested and washed in cold PBS and lysed in radioimmunoprecipitation assay (RIPA) buffer. Protein lysate was used for SDS-PAGE using 12% polyacrylamide gel. Then, proteins were transferred onto 0.1 μm nitrocellulose membrane (GE, New York, NY, USA). Free binding sites were blocked with 5% dry milk in PBS-T and treated with goat anti-His antibodies (Abcam, Waltham, MA, USA) for 1 h at room temperature. Anti-tubulin antibodies (Abcam, USA) were used for normalization. Then, secondary anti-rabbit HRP-conjugated antibodies (Abcam, USA) and chemiluminescent substrate Clarity ECL (BioRad, Hercules, CA, USA) were used to visualize proteins with the Chemidoc MP imaging system (BioRad, USA).

### 4.5. ARPE-19 Cells

The ARPE-19 cells (from the Cell Culture Collection of IDB RAS) were cultured in Dulbecco’s Modified Eagle’s Medium/F12 (1:1) supplemented with 10% fetal bovine serum. The cells were cultured at 37 °C in a 5% CO_2_ humid air atmosphere. Tissue culture dishes with a diameter of 35 mm and 6-well plates were used for culturing the cells. To induce differentiation, confluent ARPE-19 cell cultures were treated for 120 h with a concentration of 3 µM N-(4-hydroxyphenyl)retinamide (fenretinide, Sigma-Aldrich, St. Louis, MO, USA).

### 4.6. ARPE-19 Transduction and RNA Preparation

ARPE-19 cells differentiated with 3 µM fenretinide for 5 days were then transduced with either AAV9-AIPL1co, AAV9-AIPL1wt, or AAV9-GFP (100,000 viral genomes per cell (MOI)). Three biological replicates for every sample (AAV9-AIPL1co, AAV9-AIPL1wt, AAV9-GFP, non-transduced control) were used. Cells were lysed at 18 h after transduction using Lyra+ (Biolabmix, Novosibirsk, Russia). Cell lysates were frozen in liquid nitrogen and stored at −80 °C.

### 4.7. Transcriptome Sequencing

RNA extraction, libraries preparation, and sequencing were performed by LLC “Genoanalytica”. Total RNA was extracted with PureLink RNA Micro Kit according to the instructions for the kit [67]. RNA fragment analysis was performed with Agilent Bioanalyzer 2100 instrument with Bioanalyzer RNA Analysis kit [68]. RNA libraries were prepared with New England Biolabs NEBNext^®^ Ultra™ RNA II Library Prep Kit for lllumina^®^ with Poly(A) mRNA Magnetic Isolation protocol module following the instructions of manufacturer [69]. Quality control of the resulting DNA fragment libraries was performed in Agilent Bioanalyzer 2100 using the High Sensitivity Kit according to the manufacturer’s protocol [70]. Sequencing was run with the HiSeq 1500 (Illumina, San Diego, CA, USA) according to the standard Illumina protocols [71]. Over 10.1–12.8 million reads per sample were generated, and uniquely mapped reads were used for the following analysis.

### 4.8. Processing of Sequencing Results

Reads quality control, genome index building, mapping, reads-per-gene summarizing, and alignment quality control were performed on the high-throughput computing cluster “Sirius #2”. All the required software was installed with Conda 23.3.0 package manager [72]. Raw reads quality control was run on FastQC tool [73]; data summation and summary report generation were performed with MultiQC tool [74]. Genome indexes were built for reference human genome GRCh38 primary assembly [75] and v43 annotation from Gencode database [76] with the STAR tool [77] of 2.7.10b version with default parameters. Also, sequencing reads were mapped to the reference genome by STAR in quant mode “GeneCounts”. Summarizing reads by genes was performed with HTSeq-count [78] in “exon union” mode as recommended for most use cases in differential gene expression analysis. Alignment quality control was performed with the RseQC package [23]; data summation and summary report generation were performed with MultiQC tool. Subsequent analyses were performed in an integrated development environment RStudio [79] desktop version for personal computer. Counts normalization, quality control at the sample and individual gene levels, and differential expression analysis were performed using the DEseq2 package [25] of version 1.28.1 for programming language R [80]. Formula design, including only sample type, was set for DEseq2 model. Sample-level quality control using Principal Component Analysis (PCA) and hierarchical clustering methods were performed with built-in DEseq2 package functions. Pairwise comparisons between the various sampletypes were run using a negative binomial generalized linear model in DEseq2. Statistical (*p*-value adjusted = 0.005) and biological (log2FoldChange ± 1.58) thresholds were set as minimum (less conservative) at which we stopped observing changes in the number of differentially expressed genes (Appendix A). Generated lists of differentially expressed genes (with *p*-value adjusted less than 0.005 and log2FoldChange = 1.58 or higher) were run for Gene Ontology (GO) terms using GO database through application programming interface (API) function from clusterProfiler R package [81]. Visualization and illustrations were performed with ggplot2 R package [82]. To compare samples and process individual genes, intersection of gene sets was built. Lists of genes from intersections and exceptions, and also relevant gene sets from other databases, were taken for functional analysis.

## 5. Conclusions

Little is known about the regulation of *AIPL1* gene expression and its possible interactions that affect general cellular processes. This work, which presents the first RNA-sequencing of AAV9-AIPL1 transduced cells, might serve both as a primary dataset and experimental template for in-depth studies that could possibly uncover new aspects of *AIPL1*-associated biological processes. Although we did not explicitly investigate the *AIPL1* expression (Appendix A), we did observe massive changes in the expression of numerous genes in samples transduced with wild-type and codon-optimized variants; the likely relationship with *AIPL1* remains obscure. This study has defined some major pathways affected as a result of adeno-associated viral transduction. Unexpectedly, the AAV9-AIPL1 codon-optimized we developed exhibited less immunogenicity, especially of the types relevant to the target tissue. We presume that AAV9-AIPL1co influences the innate immune response and in this way affects the cellular response to interferons. This possibly facilitates the expression of transgenes. Some of these changes may highlight molecular mechanisms underlying the observed phenomena. We will seek to confirm our findings with real subretinal administration in vivo. With the presented work, we expect the assistance of further gene therapy development and applications in order to implement prospective opportunities.

## 6. Patents

*AIPL1* nucleotide sequence demonstrating high expression level confirmed by qPCR and WB is covered by patent RU 2785621.

## Figures and Tables

**Figure 1 ijms-25-00197-f001:**
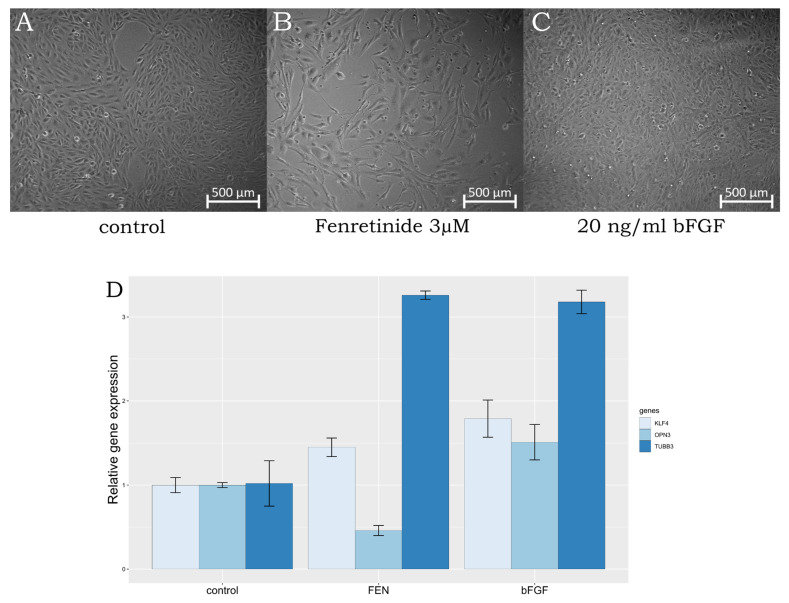
(**A**–**C**)—changes in the phenotype of ARPE-19 after 120 h of 3 µM fenretinide treatment (FEN), 20 ng/mL bFGF treatment (bFGF); (**D**)—expression of OPN3, KLF4, and TUBB3 genes treated with fenretinide, bFGF of non-treated ARPE-19 cells as control.

**Figure 2 ijms-25-00197-f002:**
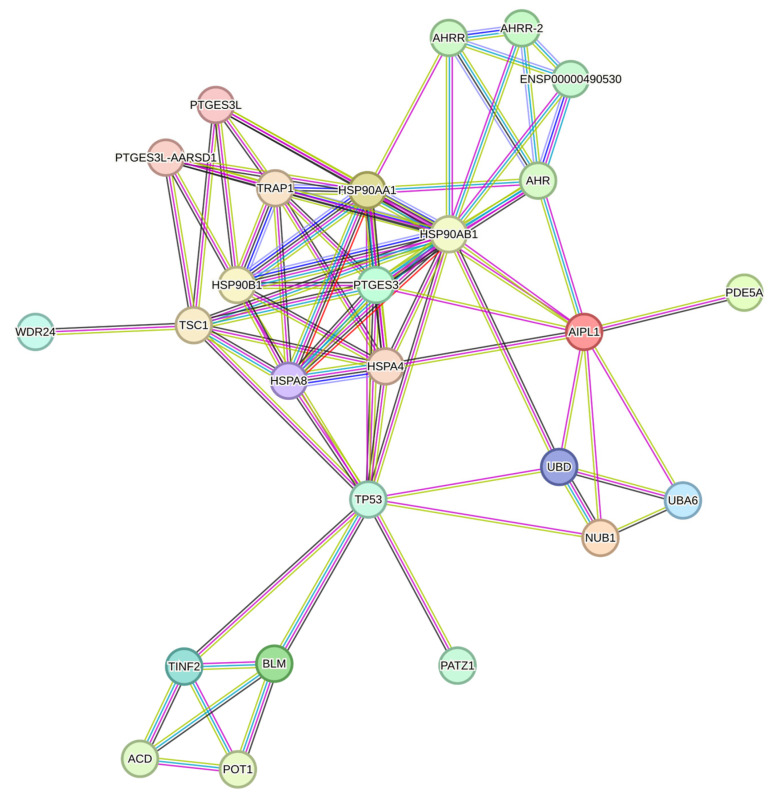
Network of proteins interacting with AIPL1 according to the STRING database.

**Figure 3 ijms-25-00197-f003:**
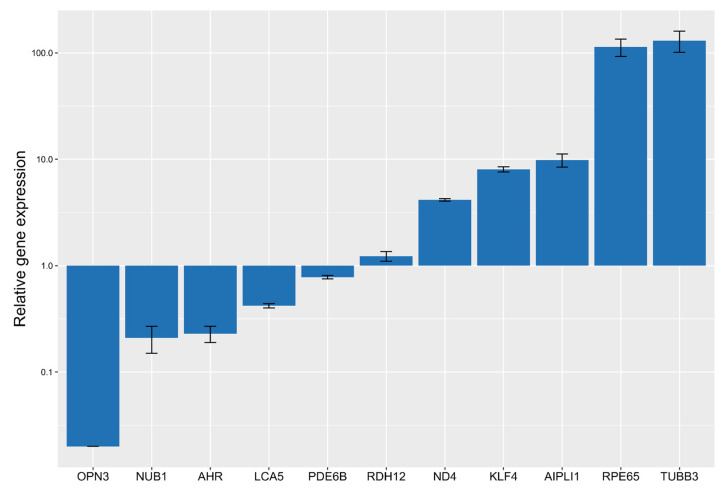
Change in the expression of genes after ARPE-19 cell trans-differentiation under the influence of 3 µM fenretinide for 5 days (square root scale).

**Figure 4 ijms-25-00197-f004:**
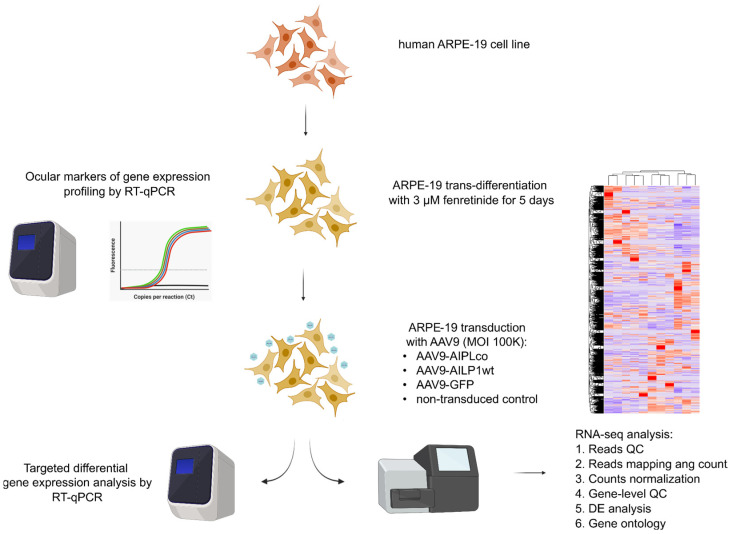
Schematic representation of the experiment with RNA-seq and subsequent data analysis of ARPE-19 cells transduced with either AAV9-AIPL1opt, AAV9-AIPL1wt, or AAV9-GFP. ARPE-19 cells were initially trans-differentiated in neuronal-like cells by incubating them with 3 μM fenretinide for 5 days. Three biological replicates were used for every sample.

**Figure 5 ijms-25-00197-f005:**
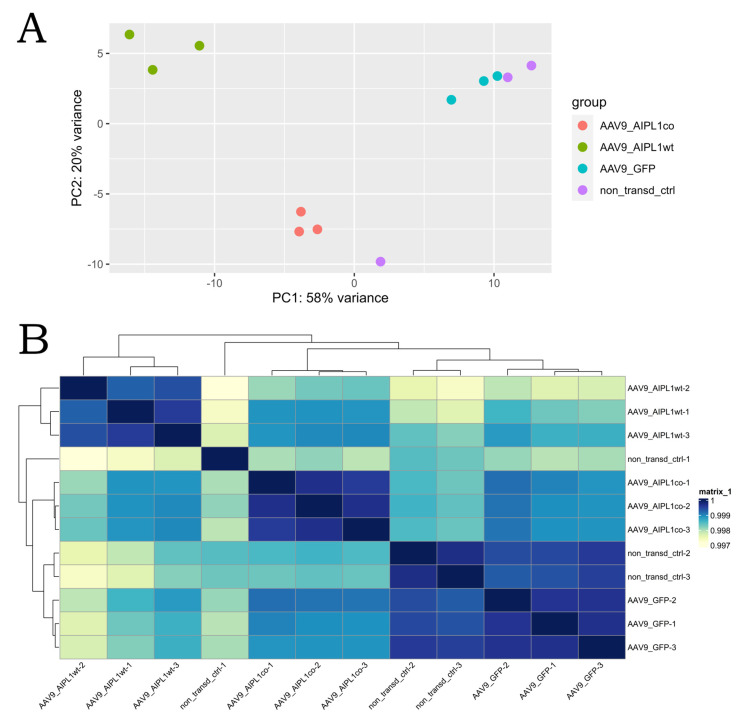
Sample level quality control results. (**A**)—PCA plot, (**B**)—hierarchical clustering heatmap. As seen from both illustrations, non-transduced control #1 does not cluster with other control replicates, and it was decided to exclude this sample from subsequent analysis to avoid excess variation between replicates.

**Figure 6 ijms-25-00197-f006:**
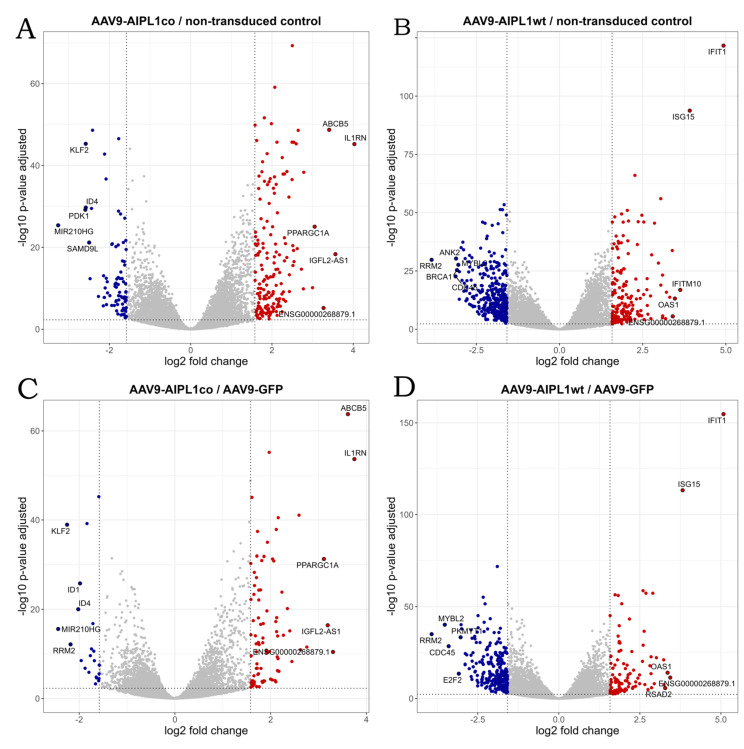
Volcano plots illustrating direction, magnitude, and significance of changes in gene expression in ARPE-19 cells transduced with (**A**) AAV9-AIPL1co vs. non-transduced control, (**B**) AAV9-AIPL1wt vs. non-transduced control, (**C**) AAV9-AIPL1co vs. AAV9-GFP, (**D**) AAV9-AIPL1wt vs. AAV9-GFP. X-axes represent log2 fold change of gene expression, Y-axes—*p*-value adjusted, computed with Benjamini–Hochberg correction. The thresholds (log2 fold change = 1.58 and *p*-value adjusted = 0.005) are marked. Positively regulated genes (red dots) and negatively regulated genes (blue dots) are shown. Top 10 genes by absolute log2 fold change in each comparison are marked with gene symbols.

**Figure 7 ijms-25-00197-f007:**
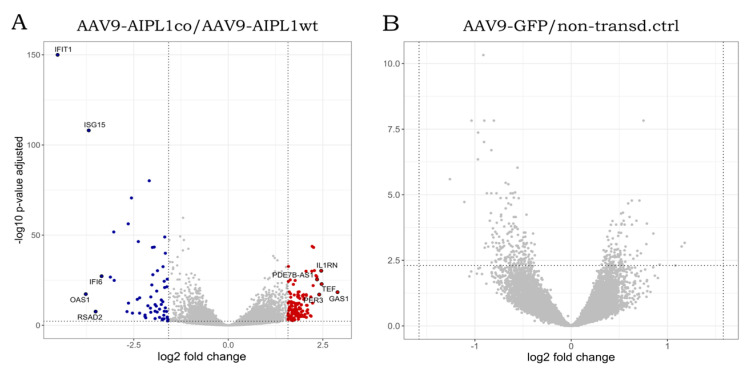
Volcano plots illustrating direction, magnitude, and significance of changes in gene expression in ARPE-19 cells transduced with (**A**) AAV9-AIPL1co vs. AAV9-AIPL1wt and (**B**) AAV9-GFP vs. non-transduced control. X-axes represent log2 fold change of gene expression, Y-axes—*p*-value adjusted, computed with Benjamini–Hochberg correction. Thresholds (log2 fold change = 1.58 and *p*-value adjusted = 0.005) are marked. Positively regulated genes (red dots) and negatively regulated genes (blue dots) are shown. Top 10 genes by absolute log2 fold change in AAV9-AIPL1co vs. AAV9-AIPL1wt comparison pair are marked with gene symbols.

**Figure 8 ijms-25-00197-f008:**
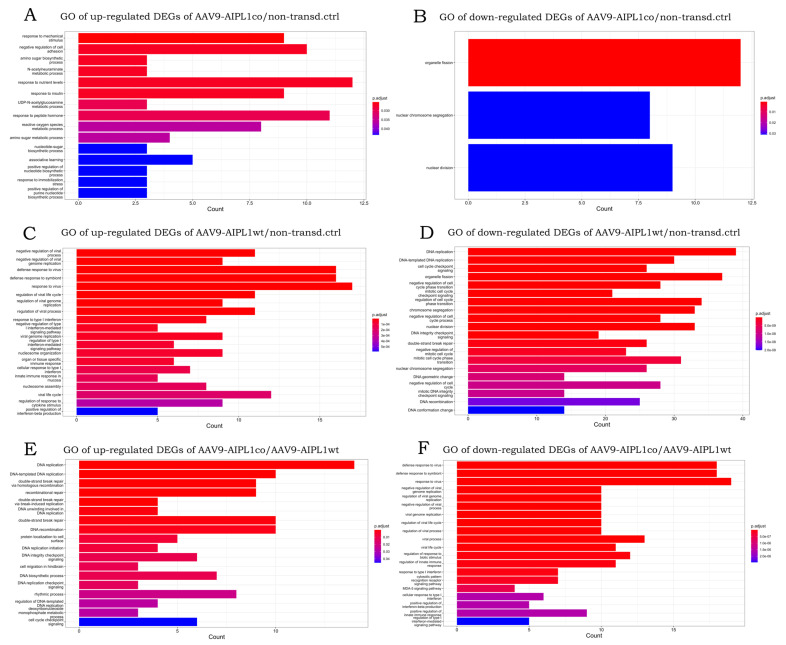
Bar plots showing top 20 Gene Ontology annotations of gene sets: (**A**)—upregulated DEGs of AAV9-AIPL1co vs. non-transduced control, (**B**)—downregulated DEGs of AAV9-AIPL1co vs. non-transduced control, (**C**)—upregulated DEGs of AAV9-AIPL1wt vs. non-transduced control, (**D**)—downregulated DEGs of AAV9-AIPL1wt vs. non-transduced control, (**E**)—upregulated DEGs AAV9-AIPL1co vs. AAV9-AIPL1wt, (**F**)—downregulated DEGs AAV9-AIPL1co vs. AAV9-AIPL1wt.

**Figure 9 ijms-25-00197-f009:**
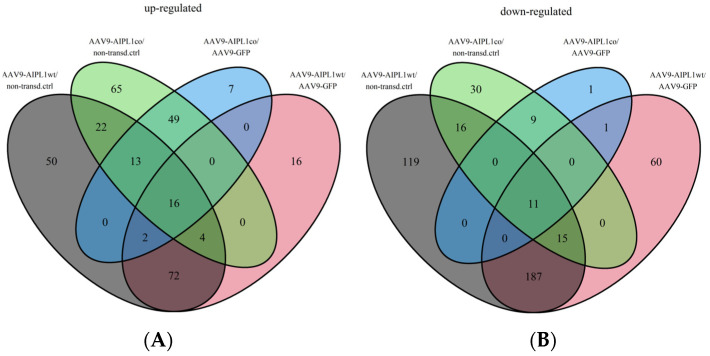
Venn diagrams representing the number of differentially expressed genes that intersect between different comparison pairs. (**A**)—intersection of all sets of upregulated differentially expressed genes, (**B**)—intersection of all sets of downregulated differentially expressed genes.

**Figure 10 ijms-25-00197-f010:**
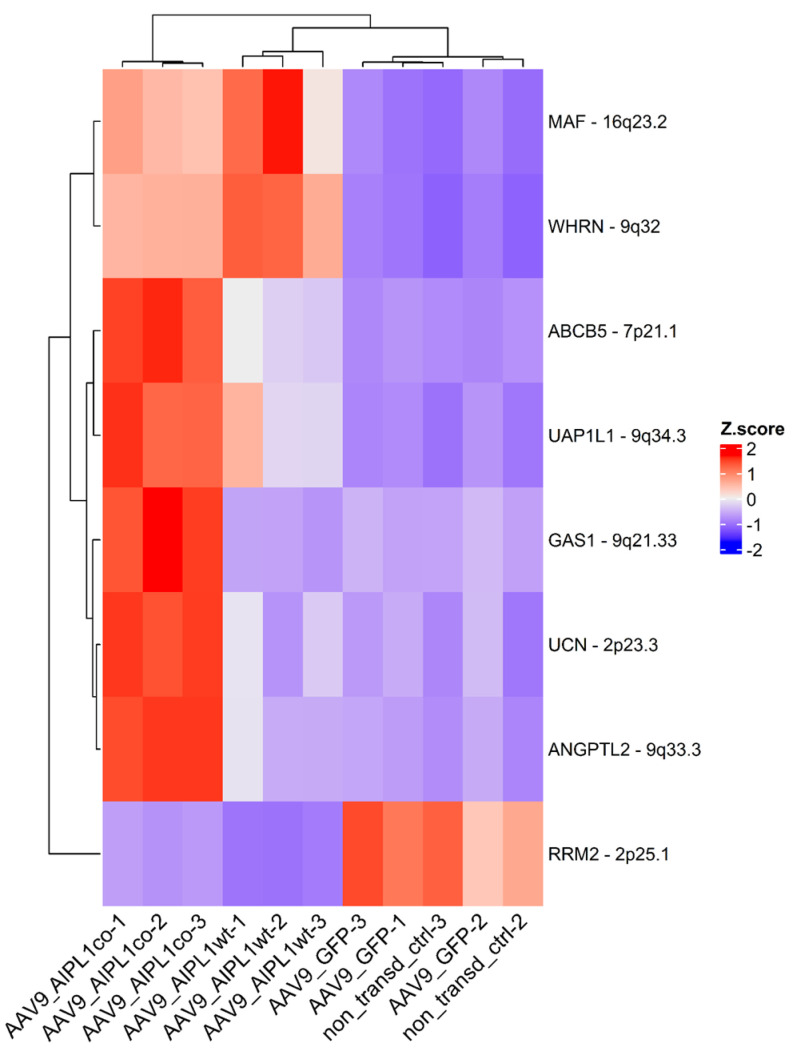
Heatmap of the most relevant genes expression change. Z-score and colors represent a scaled range of normalized read counts.

**Figure 11 ijms-25-00197-f011:**
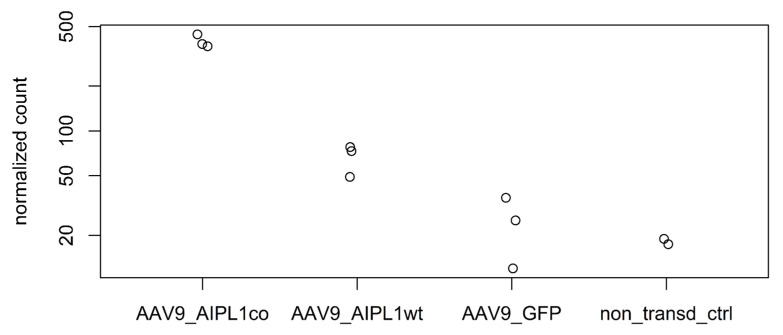
Normalized read counts of IL1RN gene in different samples.

**Figure 12 ijms-25-00197-f012:**
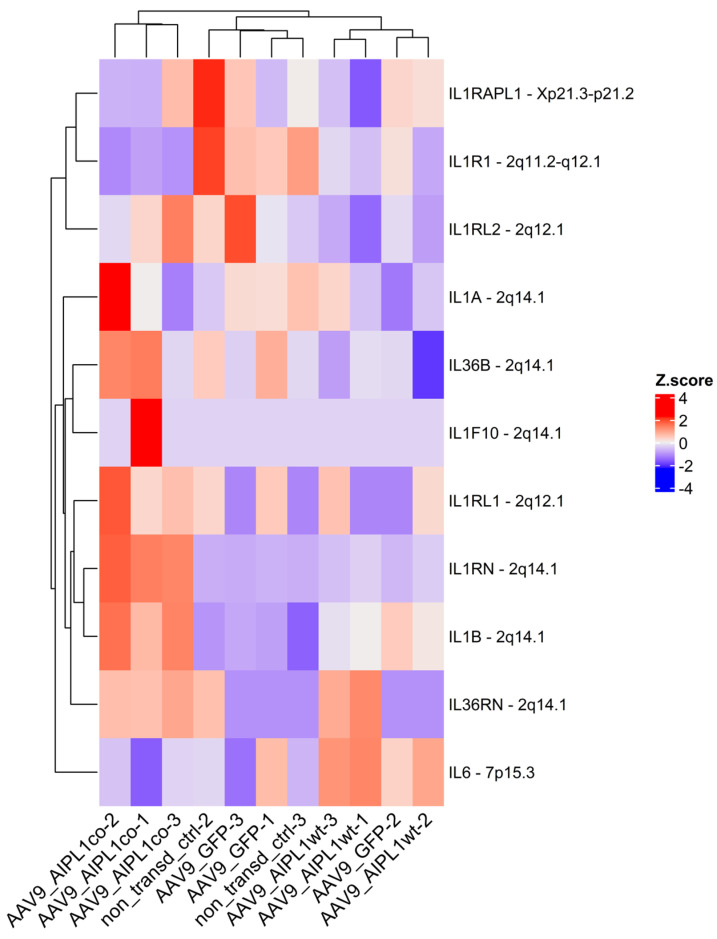
Heatmap of IL1-related genes expression. Z-score and colors represent a scaled range of normalized read counts.

**Figure 13 ijms-25-00197-f013:**
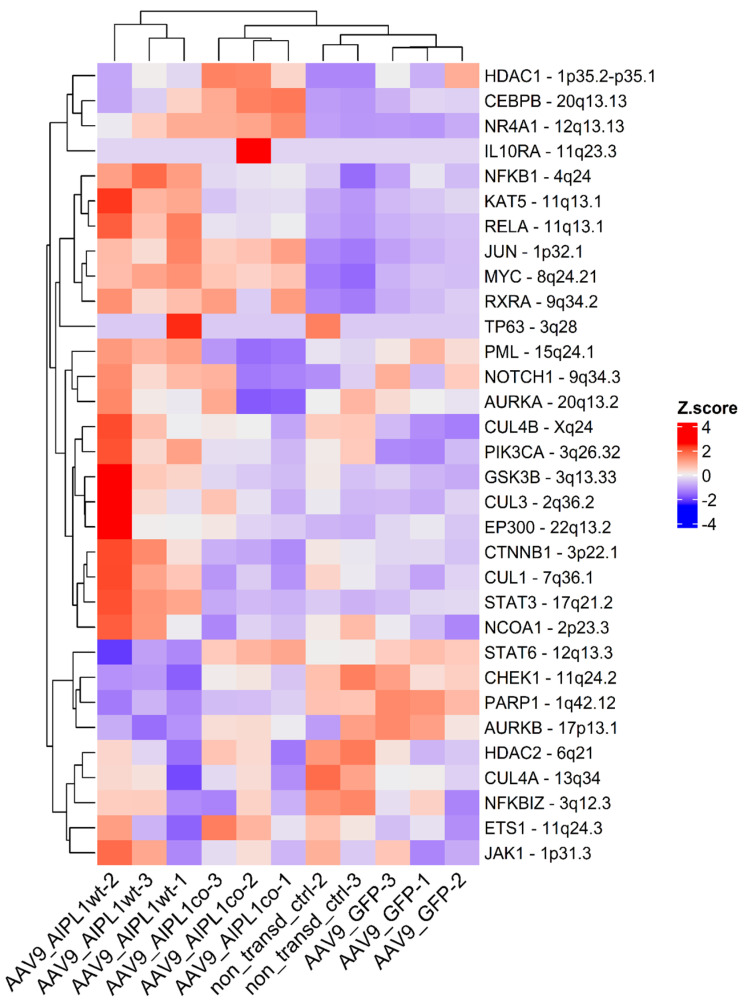
Heatmap of gene expression for NF-κB interactors. Z-score and colors represent a scaled range of normalized read counts.

**Figure 14 ijms-25-00197-f014:**
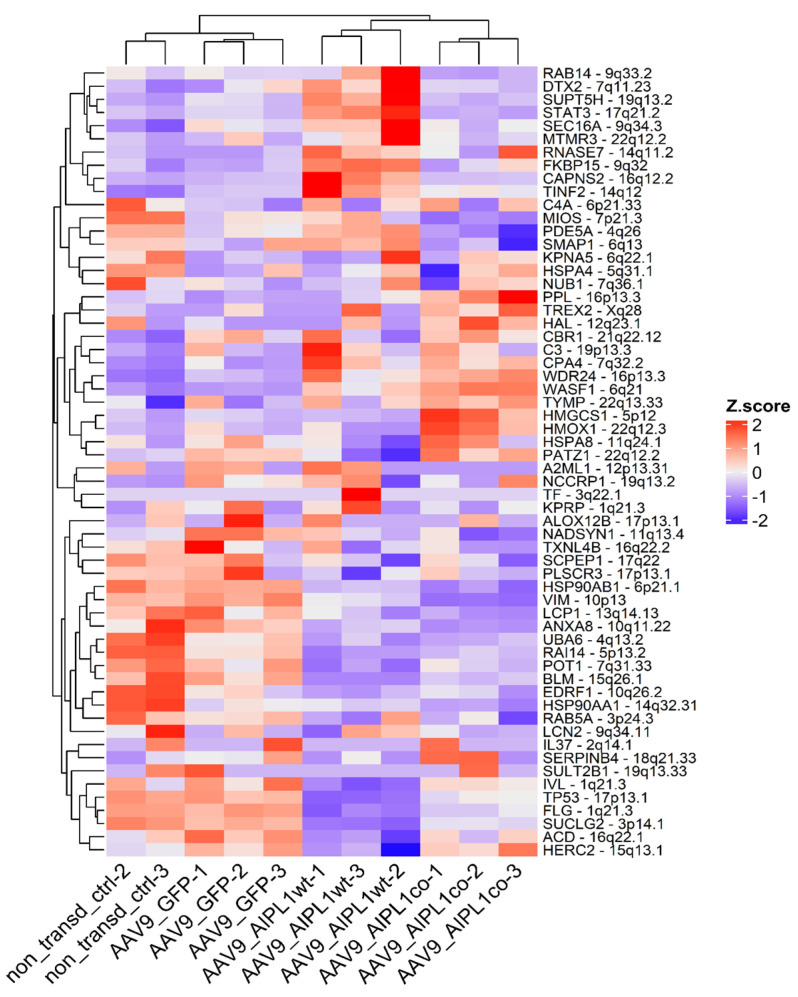
Heatmap of gene expression for AIPL1 interactors. Z-score and colors represent a scaled range of normalized read counts.

**Figure 15 ijms-25-00197-f015:**
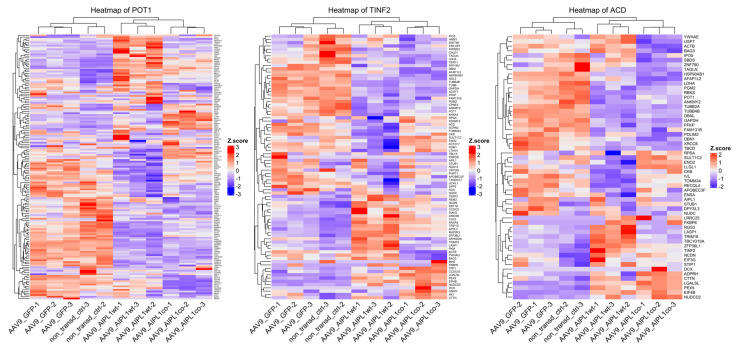
Heatmaps of gene expression for POT1, TINF2, and ACD interactors. Z-score and colors represent a scaled range of normalized read counts.

**Figure 16 ijms-25-00197-f016:**
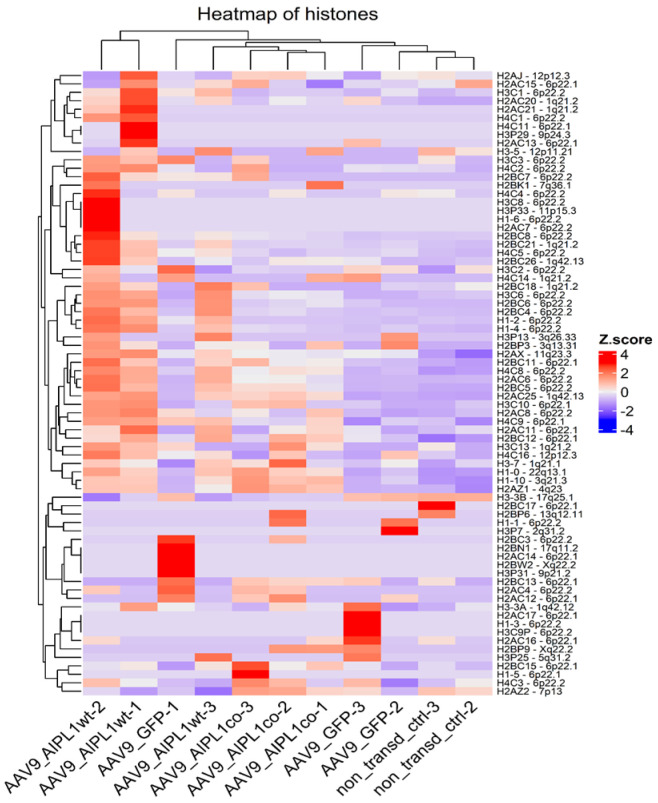
Heatmap of gene expression for histones. Z-score and colors represent a scaled range of normalized read counts.

**Figure 17 ijms-25-00197-f017:**
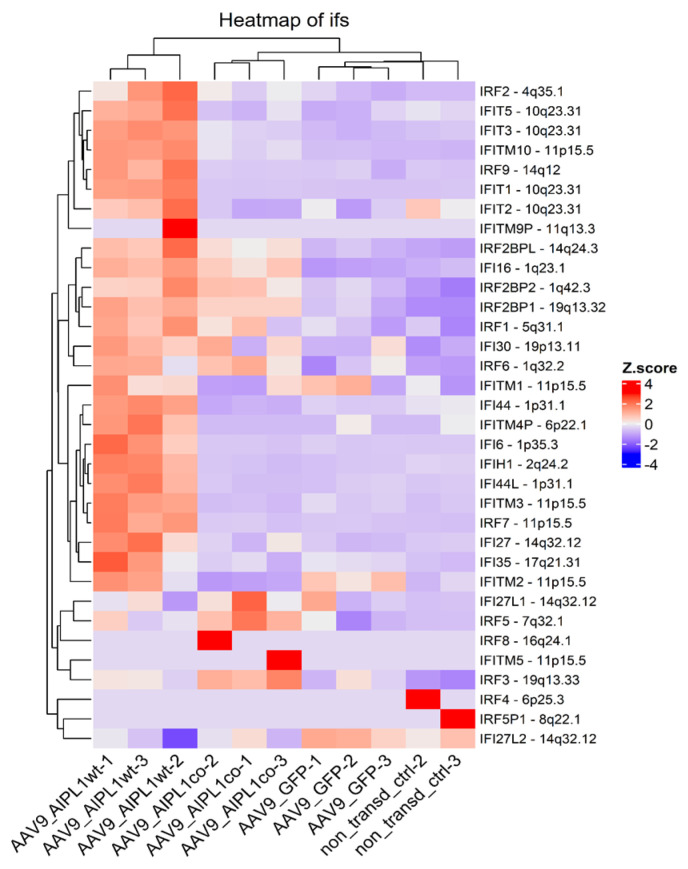
Interferon-stimulated genes heatmap, Z-score and colors represent a scaled range of normalized read counts.

**Figure 18 ijms-25-00197-f018:**
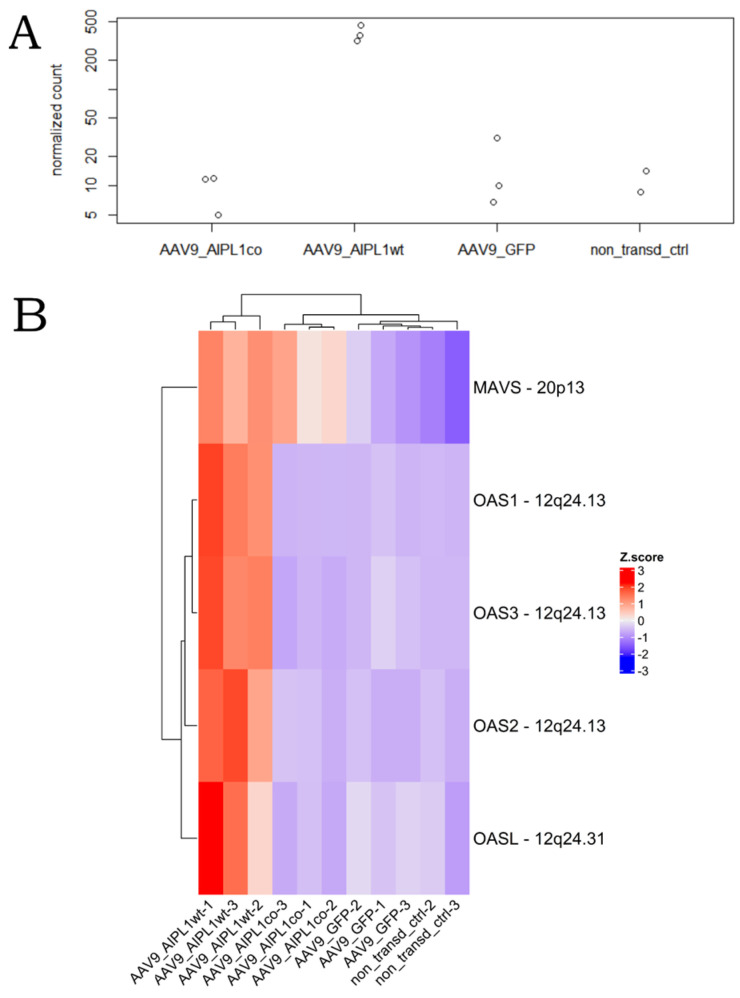
Change in OAS1 gene expression over samples. (**A**)—*OAS1* gene counts, (**B**)—dsRNA-responsive genes heatmap. Z-score and colors represent a scaled range of normalized read counts.

**Figure 19 ijms-25-00197-f019:**
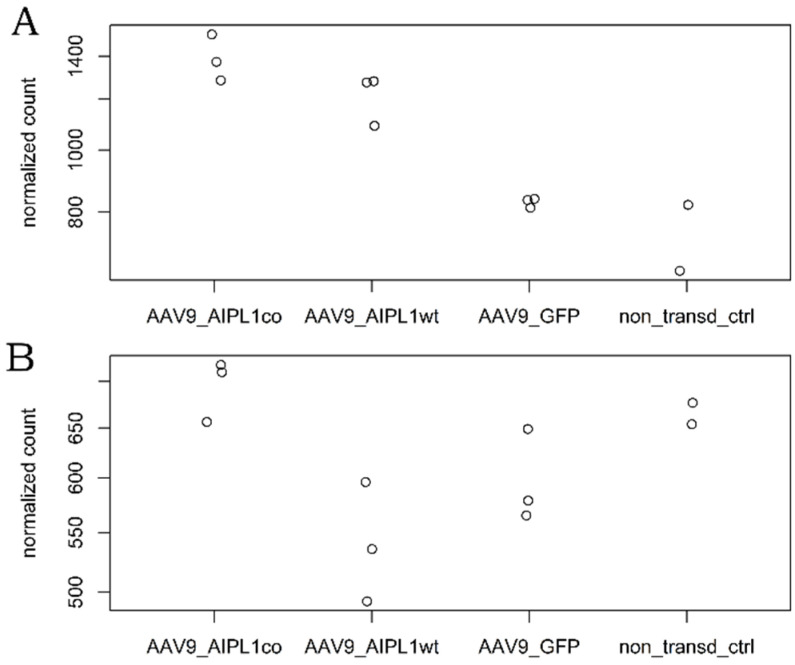
Normalized counts plot of *H2A.Z* allelic variants expression. (**A**)—*H2AZ1* gene counts, (**B**)—*H2AZ2* gene counts.

**Table 1 ijms-25-00197-t001:** Number of DEGs in different pairs of samples.

Comparison	Number of Upregulated DEGs	Number of Downregulated DEGs	Summary Number of DEGs
AAV9-AIPL1co vs. non-transduced control	169	81	250
AAV9-AIPL1wt vs. non-transduced control	179	348	527
AAV9-AIPL1co vs. AAV9-GFP	87	22	109
AAV9-AIPL1wt vs. AAV9-GFP	110	274	384
AAV9-AIPL1co vs. AAV9-AIPL1wt	129	59	188
AAV9-GFP vs. non-transduced control	0	0	0

## Data Availability

The data presented in this study are available on request from corresponding authors. The data are publicly available at https://github.com/alimagalieva/AIPL1_RNAseq_analysis.

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
