# Peer review of "RNA-Seq Analysis of Trans-Differentiated ARPE-19 Cells Transduced by AAV9-AIPL1 Vectors"

_ijms, 2023, doi:10.3390/ijms25010197_

Round 1
Reviewer 1 Report
Comments and Suggestions for Authors
The article “RNA-seq analysis of trans-differentiated ARPE-19 cells transduced by AAV9-AIPL1 vectors” describes the RNA-seq analysis of trans-differentiated ARPE-19 cells transduced with AAV9 vectors expressing codon-optimized or wt human AIPL1. AIPL1 is a photoreceptor-specific co-chaperone and mutation in the AIPL1 gene are associated to LCA4.
The authors compare RNA-seq results of ARPE-19 cells transduced with AAV9 vectors expressing codon-optimized or wt human AIPL1, or control AVV9 expressing GFP or untreated cells. The analysis shows that among the differently expressed genes, AAV9-wtAIPL1 induced the expression of interferon-stimulated genes, while AAV9-coAIPL1 was less immunogenic. The manuscript clearly describes how the data were analyzed, but a major reorganization of the manuscript is necessary for a better comprehension of the results. Many information in the Results should be moved to the “Materials and methods” session, as well as a detailed description of the figures and all their panels should be provided in the “Results” (and not in the “Discussion”), with a focus on the comparisons between AIPL1 transduced cells vs control cells.
The expression of AIPL1 in transduced ARPE-19 cells should be presented to confirm the same level of expression in both conditions. Moreover, the RNAseq data of the most relevant differently expressed genes should be confirmed by RT-PCR to further support the conclusion.
Comments on the Quality of English Language
Minor editing of English language is required.
Author Response
Dear Editors, Prof. Dr. Stylianos Michalakis Dr. Hildegard Buening! We appreciate reviewer's comments and suggestions. We are sending herewith a revised version of the manuscript entitled "RNA-seq analysis of trans-differentiated ARPE-19 cells transduced by AAV9-AIPL1 vectors". The authors are Alima Galieva, Alexander Egorov, Alexander Malogolovkin, Andrew Brovin and Alexander Karabelsky. The manuscript was carefully corrected, reformatted and improved according to the reviewers comments. We hope that our manuscript will be favorably reviewed and we are looking forward to hearing from you.
We deeply thank the reviewer for useful comments, which have greatly helped to improve our paper and better convey our ideas. Below you may find our point-by-point response to each comment.
Reviewer 1
Comments and Suggestions for Authors:
Q1: The reviewer suggests to move some details from the “Results” section to the “Materials and methods” session. In addition, the reviewer recommends to relocate all figures with captures to the “Results” session from the “Discussion”. The reorganization of the manuscript is proposed by the reviewer.
A1: We agreed with the reviewer and reformatted the manuscript accordingly. Main changes that we have made are marked in the text and all technical details regarding the RNA-seq analysis are placed in the “Materials and methods” session. Additionally, the manuscript was heavily reorganized and unnecessary details were removed with relevant references provided. All pictures accompanied with short comments have been moved to the Results section.
Q2: The reviewer kindly asks for a confirmation of the AIPL1 expression in transduced ARPE-19 cell.
A2: We agree with this comment and would like to present the data on expression of AIPL1 gene in transduced cells. Additional figure (Fig. 4) was added to the Supplementary materials 1.
Q3: The reviewer pointed out that RNAseq data of the most relevant differentially expressed genes should be confirmed by RT-PCR to further support the conclusion.
A3: We anticipate that next-generation sequencing is a high-throughput technique that allows unbiased genome-wide analyses of transcription profiles. Undoubtedly, RNA-seq has more discovery power regardless of gene sequences. Other words, RNA-seq is a hypothesis-free approach that does not require prior knowledge of sequence information. Additionally, RNA-Seq provides information about all the genes and rare transcript variants in the genome context. Moreover, computational pipelines freely available for RNA-seq data analysis are extremely robust and could be used by any researchers to validate RNA-seq data. We do believe that direct comparison of RNA-seq data with RT-qPCR is necessary only in case of low transcript coverage or significant statistical bias. Importantly, RT-qPCR results are deeply affected by assay efficiency and depend on the primers specificity to the target. In our case, AIPL1wt and AIPL1co have different sequences that require separate primers and probes for identification. We cautiously refrain from making a quantification analysis using a different set of primers for AIPL1wt and AIPL1co.
Altogether, we decided to use a discovery power of RNA-seq analysis to demonstrate any changes in the ARPE-19 cells transcription profile after AAV transduction. We also would like to share our preliminary results with the research community and will seek to ensure our findings with real subretinal administration in vivo and gene-specific RT-PCR.

Reviewer 2 Report
Comments and Suggestions for Authors
This research has tested a codon-optimized copy of the AIPL1 gene that was transferred in cultivated ARPE-19 cells by the means of a AAV9 vector. It is a very interesting work and it should be published.
I do not believe that the purpose of trans-differentiation of RPE cells (by the use of fenretinide) into expressing caracteristics for cells of neuroretinal origin has been explained sufficiently. A gene therapy engineered for the treatment of LCA type 4 should be effective in vivo, inside RPE cells situated in their normal environment and not in trans-differentiated RPE cells.
Also, the fact that the AAV containing the codon-optimized sequence of AIPL1 does not stimulate the interferon response is indeed very interesting, but it does not support the conclusion that it exhibits far less immunogenicity. After all, as the authors have highlighted, the immune response is far more complex than that and the definitive proof will come if it will be tested after real subretinal administration. The conclusion about immunogenicity should be more tempered.
Comments on the Quality of English LanguageIt is a general rule that every acronym should be explained when it appears for the first time in the text (example: wt-wild type, ER-endoplasmic reticulum, MOI-multiplicity of infection etc)
line 232 - "we suggest the insignificant level" -please rephrase.
line 248 - Klf4 -typo
line 283 - "at 48 post transduction" -do you mean 48 hours?
lines 294-295 - "the search was performed [...] allowed us" please rephrase
line 356 - "which doesn't which does not" - typo
line 373 - which is flag - please rephrase
Line 411 - resulted with list of terms - please rephrase
Author Response
Dear Editors, Prof. Dr. Stylianos Michalakis Dr. Hildegard Buening! We appreciate reviewer's comments and suggestions. We are sending herewith a revised version of the manuscript entitled "RNA-seq analysis of trans-differentiated ARPE-19 cells transduced by AAV9-AIPL1 vectors". The authors are Alima Galieva, Alexander Egorov, Alexander Malogolovkin, Andrew Brovin and Alexander Karabelsky. The manuscript was carefully corrected, reformatted and improved according to the reviewers comments. We hope that our manuscript will be favorably reviewed and we are looking forward to hearing from you.
We deeply thank the reviewer for useful comments, which have greatly helped to improve our paper and better convey our ideas. Below you may find our point-by-point response to each comment.
Reviewer 2:
Comments and Suggestions for Authors
Q1: I do not believe that the purpose of trans-differentiation of RPE cells (by the use of fenretinide) into expressing characteristics for cells of neuroretinal origin has been explained sufficiently. A gene therapy engineered for the treatment of LCA type 4 should be effective in vivo, inside RPE cells situated in their normal environment and not in trans-differentiated RPE cells.
A1: We appreciate the thorough work conducted by the reviewer, but we could not agree with the comment as the disease-causing mutation of Leber Congenital Amaurosis type 4 leads to disruption of the visual transduction process in photoreceptor cells. The characteristic phenotype of ARPE-19 cells is not identical to photoreceptors, which in their own nature exhibit neuroretinal features. Neuroretinal cells, in contrast to RPE cells, form the inner photo-sensitive layer of the eye. Neuroretina consists of six types of cells of neuronal origin (two types of photoreceptors: cones and rods, horizontal, bipolar, amacrine and ganglion cells) and three types of glial cells (Müller glial cells, astrocytes and microglial cells). Thus, with the trans-differentiation we made an attempt to create a cell model with the desired phenotype.
Q2: The fact that the AAV containing the codon-optimized sequence of AIPL1 does not stimulate the interferon response is indeed very interesting, but it does not support the conclusion that it exhibits far less immunogenicity. After all, as the authors have highlighted, the immune response is far more complex than that and the definitive proof will come if it will be tested after real subretinal administration. The conclusion about immunogenicity should be more tempered.
A2: We agree with the reviewer and attenuated the conclusion accordingly.The following text has bene edited and now in line 791 page 24:
“Unexpectedly, the AAV9-AIPL1 codon-optimized we developed exhibited less immunogenicity, especially of the types relevant to the target tissue. We presume that AAV9-AIPL1co influences the innate immune response and in this way affects the cellular response to interferons”.
Comments on the Quality of English Language
C1: It is a general rule that every acronym should be explained when it appears for the first time in the text (example: wt-wild type, ER-endoplasmic reticulum, MOI-multiplicity of infection etc)
A1: added abbreviation to every acronym. Also substituted “ER” with “endoplasmic reticulum” at line 64, “TFF” at line 110 was substituted with “tangential flow filtration”.
С2: line 232 - "we suggest the insignificant level" - please rephrase.
A2: now line 229 - “We suggest the insignificant level of expression for photoreceptor-specific genes AIPL1, LCA5, ND4, NUB1, PDE6B, RDH12 as amplification signals detected were low, indicating that the cells did not acquire basic properties of photoreceptors” rephrased to “We suggest that low amplification signals of photoreceptor-specific genes AIPL1, LCA5, ND4, NUB1, PDE6B, RDH12 indicate that the cells did not acquire basic properties of photoreceptors.”
C3: line 248 - Klf4 -typo
A3: now line 247 - corrected
C4: line 283 - "at 48 post transduction" -do you mean 48 hours?
A4: now line 280 - yes, corrected
C5: lines 294-295 - "the search was performed [...] allowed us" please rephrase
A5: now line 291 - “The STRING database search was performed to determine genes involved in AIPL1-associated biomolecular processes allowed us to generate a gene list” rephrased to “The STRING database search performed to determine genes involved in AIPL1-associated biomolecular processes allowed us to generate a gene list”
C6: line 356 - "which doesn't which does not" - typo
A6: now line 354 - corrected
C7: line 373 - which is flag - please rephrase
A7: now line 369 - “which is flag” rephrased to “which is a sign”
C8: Line 411 - resulted with list of terms - please rephrase
A8: now line 405 - “Running GeneOntology of each set of DEGs, separated to up- and down-regulated subsets, resulted with lists of terms, described on fig. 8” rephrased to “According to GeneOntology, each set of DEGs separated to up- and down-regulated subsets are distributed by biological processes described on fig. 8”
Further amendments made by the authors
line 530 - we replaced a heatmap in figure 19 with a countplot for better representation of H2A.Z genes expression
line 785 “AAV-AIPL1 transduced cell lines” corrected to “AAV9-AIPL1 transduced cells”
There were also some minor corrections in the text proposed by Reviewer 2 which we highlighted by the comments.

Round 2
Reviewer 1 Report
Comments and Suggestions for Authors
The authors have addressed most of the concerns and questions sufficiently to recommend publication of the manuscript. Some typos are still present in the manuscript. Thus, I suggest minor editing of English language.
Comments on the Quality of English LanguageMinor editing of English language is strongly suggested.